# Evaluating the Efficacy and Safety of Different Pterygium Surgeries: A Review of the Literature

**DOI:** 10.3390/ijerph191811357

**Published:** 2022-09-09

**Authors:** Marcin Palewski, Agnieszka Budnik, Joanna Konopińska

**Affiliations:** Department of Ophthalmology, Medical University of Białystok, 15-089 Bialystok, Poland

**Keywords:** pterygium surgery, pterygium, conjunctival autograft, amniotic membrane transplantation, limbal-conjunctival autograft

## Abstract

The search for the “gold standard” in the surgical treatment of pterygium has been ongoing for over two decades. Despite the development of various surgical techniques, recurrence rates range from 6.7% to 88% depending on the method used. This review discusses the latest and most commonly used methods for the surgical removal of pterygium, primarily focusing on efficacy and safety. Moreover, this review includes articles that either evaluated or compared surgical methods and clinical trials for primary and recurrent pterygium. Limited data are available on combined methods as well as on the efficacy of adjuvant treatment. The use of adjuvant intraoperative mitomycin C (MMC) and conjunctival autografting (CAU) are the two most highly recommended options, as they have the lowest rates of postoperative recurrence.

## 1. Introduction

A pterygium is a fibrovascular degenerative lesion of the conjunctiva that occurs only in humans. Clinically, it presents as a triangular-shaped conjunctival hyperplasia, with its apex directed towards the cornea. The most common location of pterygium is the inter-palpebral conjunctiva on the nasal side. A pterygium impairs vision by disturbing the tear film, inducing astigmatism, and, in severe cases, by obscuring the visual axis. Additionally, it can restrict eye movements and cause eye irritation and foreign body sensations. Changes in the local homeostasis of the ocular surface trigger the main mechanisms of pterygium formation, which include the formation of proliferative limbal stromal cell clusters, epithelial metaplasia, the formation of active fibrovascular tissue, inflammation, and the disruption of Bowman’s layer along the infiltrating apex of the pterygium [1].

Pterygium formation is one of the most common problems afflicting the anterior eye surface, with an incidence of 12% worldwide [2]. However, this varies widely by region (from 1% to 30%) [3,4,5,6]. Globally, the highest incidence rate of pterygium is found in people living between the 37th northern and southern parallels, in the so-called “pterygium belt”. The incidence of pterygium is related to sunlight exposure. Frequent and prolonged exposure to sunlight increases the risk of developing a pterygium by 24% [2]. UV-A radiation causes indirect DNA damage through the induction of reactive oxygen species and the activation of transcription factors that regulate the expression of numerous genes involved in extracellular matrix changes [7,8,9]. Discussions mainly focus on the blue component of sunlight, around 300 nm, as the cause of limbal stem cell dysfunctions. Another important point is that pterygia predominantly occur nasally. This is largely due to the Coroneo effect and deflection of the blue component of sunlight at the cornea due to a prism effect. As a consequence, sunglasses alone are not a prophylaxis, but closed temporal sunglasses that absorb the blue part of sunlight are [10]. Other risk factors include age, gender (incidence ratio for males and females = 1.3:1), smoking, having a rural background, and darker skin complexion. Genetic predisposition [11], inflammation [9], and viral infections, such as herpes simplex virus, human papilloma virus, and cytomegalovirus, also influence the development of pterygium [12].

Surgery remains the main treatment option. The main indications for pterygium removal are deterioration of vision due to pterygium enlargement, increased astigmatism, or recurrent inflammation. The aim of surgery is to remove the lesion and avoid regrowth. Surgical techniques (see Figure 1) include the bare sclera technique, conjunctival autograft, amniotic membrane graft, and conjunctival-limbal graft [13,14,15,16,17,18,19]. Treatment also includes adjuvant therapy in the form of antimetabolites, antiangiogenic agents, and radiation [20]. There are some review papers available on pterygium formation, but they describe the pathogenesis of this condition in general terms, with little emphasis on surgical treatment, and were all published in the 2010s, so are slightly dated [15,21].

To the best of our knowledge, no review papers have focused on modern techniques currently used in pterygium surgery. The aim of our study was to describe and compare the efficacy and safety of the most recent treatment methods, including adjuvant treatment.

## 2. Materials and Methods

This systematic review was conducted and reported based on the preferred reporting items for systematic reviews and meta-analyses (PRISMA) statement and the PRISMA network meta-analysis extension statement [22].

### 2.1. Literature Search

We searched PubMed, Cochrane Library, and ClinicalTrials.gov on 12 November 2021 to identify relevant studies. The following keywords were used: “pterygium”, “pterygium surgery” or “conjunctival autograft for pterygium”, and “amniotic membrane for pterygium”. We did not apply restrictions for language or date of publication.

### 2.2. Study Selection

We included articles and clinical trials that evaluated or compared surgical methods for treating primary and recurrent pterygia. Owing to the large number of results, we focused on the most frequently performed procedures. The abstracts of the articles were evaluated for relevance, followed by manually searching the results. Thus, 32 original, full-text articles were finally selected. Relevant articles were not excluded on the basis of language. Full-text translations were performed whenever necessary. We did not consider previous review papers; however, their reference list was searched for articles that met our inclusion criteria.

The following inclusion and exclusion criteria thus formed the basis for final selection for analysis.

### 2.3. Inclusion Criteria

The study was a clinical trial or randomized controlled trial.The study involved patients diagnosed with pterygium.At least one of the surgical procedures under consideration was used.The study analyzed variables such as rate of recurrence and complications.The study included an adequate follow-up period.The publication date: 2008–2021.

### 2.4. Exclusion Criteria

Case reports.Studies describing partial results (if two or more articles were based on the same cohort, we selected the paper published most recently).Duplicate publications.Review-type papers.Sufficient information was not published (e.g., full text not accessible, or full text did not contain raw data).Follow-up period shorter than 6 months.Number of participants less than 20.

### 2.5. Risk of Bias Assessment

Two authors (M.P. and J.K.) independently evaluated the methodological quality of the included studies.

### 2.6. Data Extraction

Two authors (M.P. and J.K.) independently extracted information on the types of surgical procedures, adjuvants, outcomes, and limitations. Disagreements were resolved through discussion. The PRISMA flowchart of the literature selection in this systematic review is illustrated in Figure 2 [22]. The selected articles are summarized in Table 1.

## 3. Results

### 3.1. Bare Sclera Technique

Removal of the head and base of the pterygium, leaving the sclera uncovered and cleaned, is the simplest method of surgery but is associated with an unacceptably high recurrence rate (38–88%) [42]. Regrowth occurs due to both a tendency for the proliferation of residual fibrous tissue and fibroblasts in the pterygoid locus in addition to increased inflammation within the remaining conjunctiva in the postoperative period [20]. Frequently observed complications associated with the bare sclera technique include scleral necrosis, pyogenic granuloma formation, and delayed corneal epithelialization [28].

To improve the efficacy of the bare sclera technique, adjuvant treatment in the form of mitomycin C (MMC), 5-fluorouracil (5-FU), cyclosporine, antivascular endothelial growth factor (VEGF) preparations, and irradiation has been used. Numerous studies have demonstrated the effectiveness of MMC in reducing recurrence. Cardillo et al. demonstrated that, when compared with a bare sclera technique group, the recurrence rate was significantly lower in a group of patients who intraoperatively received antimetabolites (6.66% vs. 29.97%) [43]. Subconjunctival injections performed 24 h prior to surgery were found to be as effective as intraoperative “swabs” of MMC [26]. In study conducted by Zaky et al. [26], 0.1 mL of 0.15 mg/mL mitomycin C was injected subconjunctivally into the head of the pterygium one day before surgical excision using the bare sclera technique. Following pterygium excision, intraoperative eye irrigation with 200 mL of balanced salt solution was performed to wash out residual subconjunctival MMC. Regarding postoperative complications, epithelization was delayed (more than two weeks) in two eyes (8%) in the injection group and in one eye (4%) in the group that received topical application of MMC. Scleral thinning was reported in one eye (4%) in the latter group; this occurred at one month and was resolved within three weeks under conservative treatment with topical lubricant therapy. No other serious postoperative complications were reported. MMC in the form of drops applied after treatment also significantly reduced recurrence rates from 45.5% to 10.3% [31]. In the study conducted by Hwang et al. [31], patients were randomly separated into four adjuvant therapy groups, as follows: (1) artificial eye drops four times a day for three months after surgery (control group), (2) 0.02% topical mitomycin C four times a day for five days after surgery, (3) topical 0.05% cyclosporine four times a day for three months after surgery, and (4) topical 2.5% bevacizumab delivered four times a day for three months after surgery. Authors reported significantly lower recurrence rates for the topical MMC and topical cyclosporine treatments compared with the control and bevacizumab groups. Complications associated with the use of MMC include scleral thinning, delayed corneal epithelialization (>2 weeks), and decreased endothelial cell density when used intraoperatively [44]. The number of complications increased as the duration of the use of MMC drops increased [44]. The use of 5-FU also significantly increased surgical success. Patients who had “swabs” containing 5-FU in the scleral bed had recurrence rates similar to those of CAU (11.4% vs. 12.1%). However, in eyes treated with 5-FU, there was a significant increase in postoperative complications, mainly in the form of granulomas [45]. In the study conducted by Bekibele et al., postoperative complications observed among the studied eyes included granuloma formation, at rates of 11.4% for 5-FU and 3.0% for autograft, and surface infection at a rate of 5.7% for 5-FU only. The authors used the cut conjunctiva ends, which were anchored to sclera with an 8–0 silk suture.

A significant reduction in the recurrence rate can be achieved with 0.05% cyclosporin A drops. Their use after surgery applying the bare sclera technique led to a reduction in the recurrence rate, which dropped from 45.5% to 20.6%. However, in a study by Hwang et al., cyclosporin A produced results that were inferior to MMC but superior to bevacizumab in terms of recurrence. After the removal of the pterygium, 0.05% cyclosporin A drops, 0.02% drops with MMC, and drops with bevacizumab were used. The rate of pterygium recurrence among patients was 20.6%, 10.3%, and 41.7% in the cyclosporin A, MMC, and anti-VEGF groups, respectively [31]. Cyclosporin A drops did not cause significant adverse reactions [31].

Bevacizumab was included as an adjuvant therapy due to the observed increase in levels of VEGF in the pterygium. After the intraoperative subconjunctival administration of bevacizumab, the number of recurrences was similar to that obtained using the bare sclera technique [30]. Slightly better results were observed with bevacizumab drops. The time for pterygium recurrence on the cornea was significantly longer in the adjuvant treatment group than in the group without drops, but this did not affect the recurrence rate at the end of the follow-up period [46]. Even long-term intake of bevacizumab drops did not have a statistically significant effect on the rate of corneal recurrence [42]. Both subconjunctival anti-VEGF injections and drops were well tolerated and had no serious side effects.

The application of beta radiation in a single 2500 cGy (90 Sr) dose after pterygium removal significantly reduced the incidence of recurrence compared with the bare sclera technique (11% vs. 76%). Isolated complications of abnormal eyeball adduction and mild scleral softening were observed in the group of patients that received radiation [47].

### 3.2. Removal of Pterygium with Conjunctival Autograft (CAU)

Conjunctival autograft is a newer and slightly more complicated method than the bare sclera technique for the removal of pterygia. It consists of covering the sclera, which has been cleared of Tenon’s capsules and the remnants of fibrous tissue, with a fragment of the patient’s autologous conjunctiva taken from another quadrant of either the same or the other eye. Absorbable sutures (usually 8/0 Vicryl), tissue glue (Tisseel fibrin glue), or the patient’s blood are used for graft fixation. Much better postoperative results are achieved than for removal of the pterygium using the bare sclera technique, with the recurrence rate in the analyzed studies ranging from 0% to 33.3% [13,48]. The majority of these studies have reported a recurrence rate of less than 15% for primary pterygium [13,15,18,48] when using CAU, whereas for recurrent pterygium, it is in the range of 30–33% [13,15,17]. The most common complications of pterygium removal with CAU include granuloma formation and increased intraocular pressure. However, it should be emphasized that these complications occur in isolated cases.

Sati et al. [19] found no significant difference in the recurrence rates of pterygium among different forms of conjunctival graft fixation. They obtained recurrence rates of 10% in the suture group, 6.67% in eyes with tissue glue fixation, and 3.33% in patients with autologous blood. When comparing the tissue glue method with the autologous blood procedure in a prospective randomized study, Nadarajah et al. obtained a slightly lower pterygium recurrence rate of 3.4% for the tissue glue surgery group and 10.6% for the autologous blood group at the 12-month follow up, but these differences were not statistically significant. In contrast, a study comparing suture fixation with the tissue glue method showed a higher recurrence rate when sutures were used (8.7% vs. 0%) [16]. The recurrence in the suture group was 1.4 mm onto the cornea in one patient and 2 mm for the other. It both cases, the cosmetic appearance and absence of symptoms did not necessitate further surgery. One way in which this surgical technique is inconvenient is that it takes much longer than the bare sclera technique. The use of tissue glue or autologous blood significantly reduces the procedure time compared with the sutured method (12 vs. 26 min) and decreases pain, discomfort, excessive tearing, and irritation during the first days after surgery. There was no statistically significant difference in patient-reported symptoms in later periods [16]. The most common complications associated with tissue glue or autologous blood are graft dislocation or retraction. Avoiding rubbing the eyeball after surgery reduces the risk of CAU dislocation.

The use of MMC in pterygium removal with CAU resulted in a very low recurrence rate of 0–11.8% [29,49]. Intraoperative use of MMC, along with a conjunctival graft, was shown to be a more effective method than autograft alone, with recurrence rates of 0% and 13.3%, respectively [49]. In the MMC group, “minor melting” was observed in one patient, while autograft patients had no complications. Moreover, 5-FU shows an efficacy similar to that of MMC [25,29]. In their study, Bekibele et al. found a recurrence rate of 8.7% in a group of patients treated with 5-FU and 11.8% in eyes where MMC was used. Additionally, 5-FU is also more readily available, and its use is associated with lower costs.

In summary, autologous blood for conjunctival autograft fixation is associated with lower graft stability. Therefore, graft retraction and displacement have been more predominantly applied in autologous blood groups than in fibrin glue and suture groups. However, the use of autologous blood did not result in higher recurrence rates than the use of fibrin glue or sutures. The main advantage of autologous blood is that it is more readily available and less expensive compared with other methods of fixation. Moreover, patient satisfaction is higher and there are less severe postoperative symptoms in the blood coagulum group than in the groups treated using the other techniques [50].

### 3.3. Removal of Pterygium with Amniotic Membrane Transplantation (AMT)

The desire to preserve the conjunctiva has led to the use of an amniotic membrane to cover the scleral bed. An amniotic membrane consists of a single layer of epithelium with a basement membrane and an extracellular matrix (ECM) layer. The ECM consists of a compact cell-free layer and a loose fibroblast layer [51]. Owing to its biological properties, the amniotic membrane can be used as a graft with anti-inflammatory and antifibrotic properties that can deliver numerous growth factors and promote epithelial cell proliferation and differentiation without the risk of immunological reactions [52].

Reported recurrence rates among patients after AMT surgery range from 6.7% to 40.9% [36,53]. Such variation in the recurrence rate is a result of various factors, such as differences in the surgical procedures performed (such as the use of fibrin glue and sutureless procedures) and the characteristics of the study population (such as sex distribution and the mean age of participants); moreover, differences in the definition of recurrence also appear to contribute, which vary according to the surgical method. In addition, the use of amniotic membranes is associated with a lower recurrence rate than the bare sclera technique (12.9% vs. 37.5%) [54]. Published data differ regarding whether AMT or CAU is indicated to be more effective. Most of the papers analyzed here show that amniotic membrane transplantation yields higher rates of recurrence than conjunctival autotransplantation [24,55,56,57,58,59,60,61]. These differences are significant in both primary and recurrent cases. Prabhasawat et al. achieved a recurrence rate of 10.9% in primary pterygium and 37.5% in recurrent pterygium in eyes with AMT, while the rates in patients with CAU were 2.6% and 9.1%, respectively [62]. In comparison, Cem et al. reported no statistically significant differences in recurrence rate between CAU and AMT for both primary and recurrent pterygium [63].

The addition of MMC to AMT did not affect the recurrence rate [64]. Additionally, Yasemin et al. showed that pterygium recurrence rates were similar whether MMC was used with an amniotic membrane or with CAU [33].

Complications of AMT appear very rare but may eventually occur. Such reported complications include double vision, granuloma, ocular motility disturbances, and symblepharon. Amniotic membrane transplant, being allogeneic, carries the potential risk of transmitting infectious diseases. However, this has not been reported in any of the studies we analyzed.

### 3.4. Removal of Pterygium with Limbal Conjunctival Autograft (LCAG)

The conjunctival-limbal grafting method was invented to prevent potential limbal cell deficiency at the graft site after pterygium removal [65,66]. This method involves transplanting a fragment of the patient’s conjunctiva with a fragment of the limbus, which contains stem cells, to the site of the removed pterygium; this may facilitate faster epithelialization of the corneal epithelium [67]. As with the previous methods, it is possible to attach the graft using both sutures and tissue glue. This procedure has been shown to be effective, with recurrence rates ranging from 0% to 18% [13,53,68,69]. A study comparing CAU with LCAG in eyes after pterygium removal and the application of MMC at 0.02% for 3 min showed no statistically significant difference in postoperative recurrence (0% in LCAG and 5.1% in CAU) [70]. However, in the LCAG group, 12.8% of patients had a concomitant occurrence of neovascularization at the site from which the horn graft was harvested, while patients with CAU were free of such complications. Chen et al. compared LCAG and AMT in the treatment of recurrent pterygium. Both groups were additionally treated with intraoperative MMC soaks at a concentration of 0.02% for 3 min. At the end of the 12-month follow-up period, the differences in recurrence rates were not statistically significant, with 2.1% in the LCAG + MMC method and 10.9% in AMT + MMC method [34].

Postoperative application of cyclosporin A drops applied at a concentration of 0.05% to patients undergoing LCAG results in a reduced recurrence rate (from 17.9% to 3.4%) [71]. In addition, cyclosporin A reduced postoperative pain. In a study by Aydin et al., the mean visual analog scale (VAS) pain score in the first week after surgery was significantly lower in the cyclosporin A group [71].

The most common complications of LCAG include pannus at the donor site, conjunctival hyperemia, subconjunctival hemorrhage, conjunctival epithelial defects, and graft retraction [38]. Conjunctival cysts, symblepharon, and postoperative wound dehiscence are less frequently observed [23,72].

## 4. Discussion

Pterygium formation is a common disorder of the anterior eyeball segment. Its prevalence reaches 22% in equatorial areas and is less than 2% in latitudes above 40° [73,74]. Postoperative recurrence is a significant complication. By analyzing the effectiveness of the surgical methods selected for our study, we found that the technique with the highest recurrence rate was the bare sclera technique. Therefore, this method should no longer be used. Better postoperative results can be obtained by combining the bare scleral technique with adjuvant therapy. MMC and 5-FU are widely available therapies that have proven efficacy in reducing the rate of recurrence. However, it has been shown that MMC used intraoperatively, at a concentration of 0.04%, and in combination with bare sclera is less effective than surgery with CAU (38% vs. 15%) [17]. In addition, the number of complications was significantly higher in the MMC group. When 5-FU + bare sclera was used [45], rates of pterygium recurrence similar to those in eyes with CAU were achieved; however, CAU again proved to be the safer procedure. Thus, autoconjunctival transplantation allows for similar or better efficacy than the bare sclera technique with adjuvants while offering the possibility of avoiding complications, including serious, and vision-threatening complications associated with the use of antimetabolites. Currently, this is the method of choice for primary pterygium surgery.

The choice of fixation method is of secondary importance, as studies have shown no significant differences in recurrence rates, regardless of the method of fixation. In developing countries, many of which are in the pterygium belt, the patient’s blood is frequently used for fixation. Compared to sutures, this significantly reduces the procedure time. Although lower graft stability was observed with blood fixation than with tissue glue [74], it is not frequently used due to its high cost (USD 140) and low availability in the countries that have the highest prevalence of pterygium.

Previous studies have reported that the efficacy of CAU is similar to that of LCAG for primary pterygium; however, the LCAG method carries the risk of corneal neovascularization at the site of autograft retrieval, which is not observed in CAU [41]. LCAG is also a longer and more technically difficult surgical method than CAU.

Removal of the pterygium with a conjunctival graft of the patient’s conjunctiva is currently the recommended method. However, this method has its limitations. Firstly, it cannot be used in eyes with extensive pterygium, double-headed pterygium, post-inflammatory lesions of the conjunctiva, or in those scheduled for future antiglaucoma surgery. An alternative is AMT, which is a safe method with few intraoperative and postoperative complications. The success rate of surgery with amniotic membranes is much higher than for pterygium removal using the bare sclera technique; however, according to most researchers, it is lower than that of surgery with CAU. Adding an antimetabolite to surgery with AMT achieves a recurrence rate comparable to that of CAU therapy with MMC (8% AMT + MMC vs. 13.3% CAU + MMC) [33].

The second group of patients for whom CAU may not be the optimal treatment method for pterygium removal consists of those with a higher initial risk of recurrence, including patients under 50 years of age [41] and with a pterygium that is fleshy [75] or has exacerbated inflammation [76]. The addition of MMC to CAU reduces the recurrence rate compared with CAU (0% vs. 13.3%) [49]. Intraoperative or postoperative use of this antimetabolite allows for a recurrence rate of <4% [37,49]. Application of MMC for 3 min at a concentration of 0.2 mg/mL during surgery is considered the safest approach [49]. Thus, we believe that the addition of low-dose MMC to CAU is a good therapeutic option for patients at high risk of pterygium recurrence.

Another important issue related to success rates is the use of β-radiation (RT) as an efficient adjuvant application. It prevents revascularization, which damages the endothelial cells lining the lumen of capillaries [77]. Postoperative RT is an efficacious adjuvant therapy used to prevent pterygium recurrence, and it has been shown to significantly inhibit pterygium regrowth. However, a consensus has not yet been reached regarding the optimal β-radiation dose for pterygium recurrence. Although it has been suggested that 30–50 Gy is the required dose range from a radiation oncology perspective [78], a high incidence of scleromalacia has been reported after RT. RT alone produces a significant reduction in the size of the pterygium [20,23,57]. However, this is usually insufficient for achieving a complete cure, perhaps because it acts on relatively mature tissues, which are less affected by irradiation. For better results, RT should be applied after surgery. The procedure can be performed from a few hours after surgery to a few days after surgery; it can be repeated within 24 h after surgery, and then at 1 week postoperatively.

Different doses of β-radiation may cause different success rates. In the study undertaken by Yamada et al. [79], β-radiation was more efficacious at 40 Gy than at 20 Gy in preventing pterygium recurrence without the development of scleromalacia, particularly for large-size pterygia and for those in young patients.

Our study has several limitations. The analyzed studies differ in terms of the inclusion criteria and group sizes. In addition, not all of the studies offer a precise definition of pterygium nor include an assessment of either disease progression or recurrence rate. Furthermore, the follow-up period after recurrence varies between studies.

## 5. Conclusions

Although exposure to UV light remains a major risk factor for the development of pterygium, other factors, such as viral infections and genetic susceptibility, cannot be ignored. As the literature lacks reliable conclusions and clear evidence, these hypotheses have little to no effect on ophthalmology practice.

Adjuvant intraoperative MMC and conjunctiva autografting are the two most highly recommended procedures, as they have the lowest rates of postoperative recurrence. In addition, the application of cyclosporin A at the site of the excised pterygium has been proven to be safe. However, it is difficult to assess the efficacy of these adjuvant treatments, as the reported data are still inconclusive.

Postoperative recurrence is a significant problem in ophthalmic practice. Additional studies are needed to improve the existing techniques and to investigate newer surgical techniques. Conjunctival autograft is one of the surgical techniques for which the effectiveness and long-term effects should be investigated further. There are also sparse data available concerning the use of the conjunctival-limbal graft method in combination with the simultaneous covering of the defect with an amniotic membrane.

Furthermore, additional meta-analyses comparing different surgical methods are needed.

In conclusion, we presented data regarding the efficacy and safety profiles of the pterygium surgery techniques that are most commonly used worldwide. CAU has a lower recurrence rate compared to that of the bare sclera technique and AMT in addition to having a better safety profile compared to techniques augmented with antimetabolites; accordingly, CAU remains the first-choice method for primary pterygium surgery. AMT with MMC is a good alternative for patients with recurrent pterygium. In selecting a “tailored” method from the spectrum of available surgical techniques for pterygium surgery, surgeons should consider the patient’s needs and their past and anticipated surgeries as well as the local eye conditions.

## Figures and Tables

**Figure 1 ijerph-19-11357-f001:**
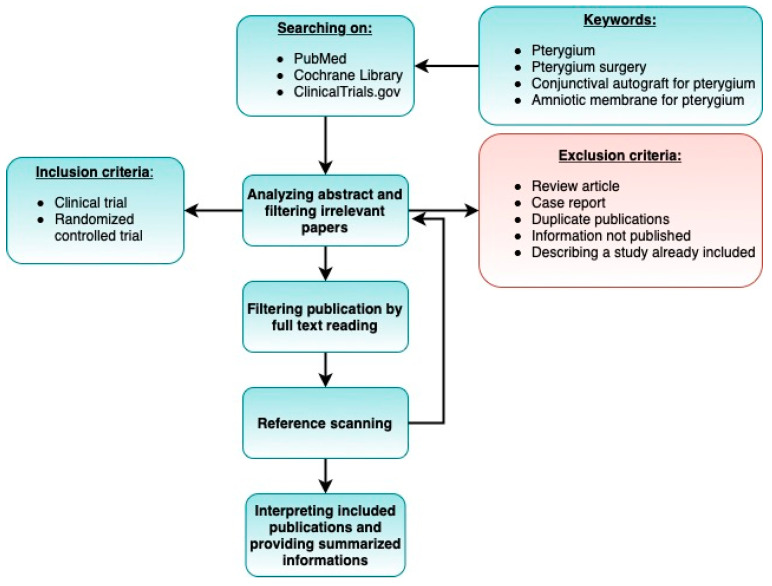
Search strategy.

**Figure 2 ijerph-19-11357-f002:**
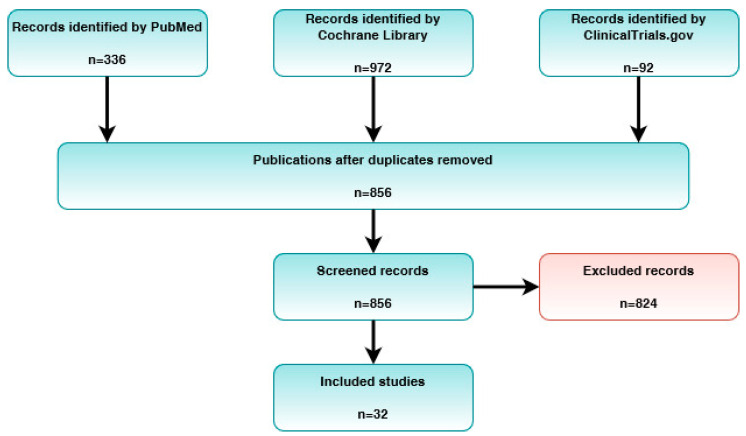
PRISMA flowchart of the study selection process.

**Table 1 ijerph-19-11357-t001:** Characteristics of the clinical studies included in the analysis.

Reference and Date	Participants (Number)	Aim of Study	Follow Up Period (Months)	Preoperative Assessment 1	Postoperative Treatment	Recurrence Rate (%) 2	Safety
**Jha KN** **(2008)** [23]	32	LCAG: primary vs. recurrent pterygium	18	L: 93.75% N/6.25% T M: no data S: 2–4 mm	Gutt ATB + GCs qid, 2 w. Sutures removal: YES (14 d)	Primary pterygium—0% Recurrent pterygium—0%	Conjunctiva cyst in 2 patients Non-major postoperative complications
**Reece C Hall** **Andre J Logan** **(2009)** [16]	50	CAU + 8/0 Vicryl vs. CAU + fibrin glue	12	L: 100% N M: no data S: >4 mm	Gutt ATB qid 1 w; gutt DEX qid 1 m. Sutures removal: NO	Vicryl group—8.7% Fibrin glue group—0%	Fibrin glue group: Graft dehiscence in 2 patients Nasal retraction of the graft in 8 patients Granulomas in 3 patients
**Wanhong Liang** **Rongxu Li** **(2012)** [24]	133	CAU vs. AMT	12	L: 100% N M: no data S: pterygium invaded pupil	Gutt ATB qid 1 w, ung ATB od 1 w. Sutures removal: YES (12 d)	CAU—7.4% AMT—19.2%	Foreign body sensation or discomfort: CAU group—13.58% AMT group—32.63% Eyelid edema and conjunctival hyperemia edema: CAU group—9.88% AMT group—23.08%
**Charles O Bekibele** **Adeyinka Ashaye** **(2012)** [25]	88	CAU + 5-FU vs. CAU + MMC	8	L: 5-FU: 95.65% N/4.35% T MMC: 95.6% N/4.4% T M: (T1/T2/T3) 5-FU:2.2%/6.5%/91.3% MMC:0%/0%/100% S: 5-FU: 3.4 ± 1.3 MMC: 3.3 ± 1.2	Ung ATB tid, gutt DEX qid 6–10 w. Wearing dark glasses advised.	CAU + 5-FU—8.7% CAU + MMC—11.8%	Corneoscleral melting MMC group—2.9% Sclera granulom–5-FU group—6.5% MMC group—2.9% Graft rejection-5-FU group—6.5% MMC group—2.9% Symblepharon-5-FU group—2.2% MMC group—2.9%
**Khaled A Zaky, Yasser M Khalifa** **(2012)** [26]	50	BS + subconj. injection MMC (0.1 mL of 0.15 mg/mL) the day before surgery vs. BS + intraoperative topical application of MMC (0.1 mL of 0.15 mg/mL)	12	L: no data M: no data S: 3 to 5 mm	Gutt ATB qid 4 w, gutt DEX qid 4 w, ung ATB + DEX od 4 w.	0% in both groups	Delayed epithelization: subconj. MMC group—8% MMC application group—4% Scleral thinning: MMC application group—4%
**Gabor Koranyi Ditte Artze’n** **(2012)** [27]	115	BS + intraoperative 0.04% MMC vs. CAU + 7/0 Vicryl	48	L: no data M: no data Area: MMC group: 6.5 mm 2 ± 4.0 CAU: 7.5 mm 2 ± 5.5	Gutt DEX 6/d 1 w tapered-off over the next 5 w, ung ATB tid 1 w. Sutures removal: no data.	MMC group—38% CAU group—15%, (*p* < 0.05).	Graft necrosis with subsequent keratitis: CAU group—1 patient Postoperative scleral melting: MMC group—1 patient Delayed epithelialization: MMC group—16 patients Mild scleral thinning: MMC group—8 patients CAU group—2 patients
**Kheirkhah A,** **Nazari R** **(2013)** [28]	54	BS + intraoperative MMC and TMC vs. BS + intraoperative MMC	12	L: 100% N M: TMC group: 17.4%/56.5%/26.1% Control group: 20%/56%/24% Size: no data	Gutt ATB 2 w, gutt GCs (BTM→FML) 3 m	Conjunctival recurrence rate: TMC group—8.7% Control group—4.0%, (*p* < 0.05). Corneal recurrence rate: 0% in both groups.	Conjunctival inflammation Pyogenic granuloma Increased intraocular pressure in steroid group
**Alok Sati,** **Sandeep Shankar** **(2014)** [19]	90	CAU + 8/0 Vicryl vs. CAU + fibrin glue vs. CAU + autologous blood	12	L: no data M: Vicryl: 20%/53.33%26.67% Fibrin glue: 20%/53.33%26.67% Autologous blood: 20%/53.33%26.67% S: Vicryl: 3.73 ± 0.41 Fibrin glue: 3.93 ± 0.5 Autologous blood: 3.85 ± 0.42	Gutt DEX qid 1 m, ung ATB qid 1 w. Sutures removal—no data	Vicryl group—10% Fibrin glue group—6.67% Autologous blood group—3.33%	Autologous blood group: Graft retraction—10% Graft displacement—6.67% Conjunctival cyst—3.33%
**Olusanya BA,** **Ogun OA** **(2014)** [29]	80	CAU + MMC vs. CAU + 5-FU	8	L: 92.4% N/1.3% T/6.3% D M: 1.3%/3.7%/95% S: no data	Ung ATB qid, gutt DEX qid, 6–10 w.	MMC group—11.8% 5-FU group—8.7%	No data
**Julia Promesberger,****Sharmila Kohli** **(2014)** [30]	81	Primary pterygium: CAU (P1) vs. CAU + PTK (P2) Recurrent pterygium: CAU (R1) vs. CAU + PTK(R2)	>24	Primary pterygium: L (P1 + P2): 87.7% N/3.5% T/8.8% D M: no data S: no data Recurrent pterygium: L (R1 + R2): 87.1% N/3.2% T/9.7% D M: no data S: no data	Gutt ATB qid, P1 + R1: dexpanthenol cream qid P2 + R2 gutt NSAID 5x/d.	P1—32.5% P2—7.1% R1—26.3% R2—23.1%	No data
**Shinyoung Hwang,** **Sangkyoung Choi** **(2015)** [31]	132	BS + POV vs. BS + 0.02% MMC vs. BS + 0.05% CYP vs. BS + 2.5% BEV	6	L: no data M: POV: 9.1%/36.4%/54.5% MMC: 13.8%/37.9%/48.3% CYP: 14.7%/26.5%/58.8% BEV: 11.1%/27.8%/61.1% S: POV: 3.7 ± 0.5 MMC: 3.8 ± 0.7 CYP: 4.1 ± 0.3 BEV: 3.4 ± 0.9	Gutt ATB qid 1 w, FML qid 1 m utt: POV, 0.02%MMC, 0.05% CYP, 2.5% BEV qid 3 m.	POV—45.5% MMC—10.3% CYP—20.6% BEV—41.7%	Subconjunctival hemorrhage: POV group—2 patients MMC group—1 patient CYP group—1 patient BEV group—2 patient
**Ngamjit Kasetsuwan,** **Usanee Reinprayoon** **(2015)** [32]	22	BS vs. BS + 0.05% BEV	6	L: no data M: BS: 0%/75%/25% BS + BEV: 10%/70%/20% S: no data	Gutt ATB + DEX qid 1 m, Gutt FML qid 2 m. BS: gutt AT qid 3 m BS + BEV: gutt 0.05% BEV qid 3 m.	BS—90% BS + BEV—33.33%	Conjunctival defect: BEV group—1 patient Corneal epithelial defect: BEV group—1 patient
**Yasemin Arslan Katircioglu,** **Ugur Altiparmak** **(2015)** [33]	60	AMT + MMC vs. CAU + MMC	27	L: no data M: no data Size: no data	Gutt ATB qid 1 w, gutt AT qid 1 w, gutt PRED qid 1 m. Sutures removal: YES (within 14 d)	AMT + MMC—8% CAU + MMC—13.3%	Severe pain: AMT + MMC—8% CAU + MMC—13.4% Epithelial defect AMT + MMC—20% CAU + MMC—13.4%
**Rongxin Chen,** **Goufu Huang** **(2016)** [34]	96	LCAG + intraoperative 0.02% MMC vs. AMT + intraoperative 0.02% MMC	12	L: LCAG group: 89.4% N/10.6% T AMT group 86.9% N/13.1% T M: no data S: LCAG: 3.77 ± 1.38 AMT: 4.19 ± 1.88 (*p* < 0.05)	Gutt ATB, gutt AT, ung ATB; gutt NSAID at the end of the first week, gutt + ung ATB + DEX 1 m. Sutures removal: YES (14 d).	LCAG group—2.1% AMG group—10.9%	Local episcleral melting with graft melting in: LCAG group—2 eyes AMT group—1 eye Localized pannus formation at the donor site: LCAG group—5 eyes
**Gaayathri Nadarajah,** **Vanithe H Ratnalingam** **(2017)** [35]	120	CAU + autologous blood vs. CAU + fibrin glue	12	L: no data M: Autologous blood group: 11.7%/25%/15% Fibrin glue group: 10.8%/25.8%.11.7% S: no data	Gutt ATB + DEX q 2 h 1 w, qid 3 w, bds 2 w. Ung ATB od 6 w.	Autologous blood group—10.6% Fibrin glue group—3.4%	Partial graft dislodgement: Autologous blood group—9.7% Fibrin glue group—1.7% Complete graft dislodgement: Autologous blood group—24.2%
**Mitra Akbari,** **Reza Soltani-Moghadam** **(2017)** [36]	60	AMT vs. CAU	12	L: no data M: no data S: no data	Gutt ATB 2 w Gutt BTM 3 m	AMT group—6.7% CAU group—3.3%	Higher inflammation in the AMT group. No severe events in both groups
**Ayman Lotfy,** **Ahmed A. M. Gad** **(2018)** [37]	108	CAU + preoperative MMC injection vs. CAU + intraoperative MMC Group I—injection of 0.1 mL of 0.15 mg/mL MMC into the body of pterygium 1 day before surgery Group II—local application of 0.2 mg/mL MMC for 2 min over the medial rectus tendon during surgery.	23	L: no data M: no data S: Group I: 3.93 ± 0.65 Group II: 3.91 ± 0.6	Gutt PRED qid 1.5 m, gutt ATB qid 1 w. Sutures removal: YES (3–4 w)	Group I—3.92% Group II—1.85%	Dellen: Group I—1 eye Conjunctival cyst: Group II—1 eye Conjunctival granuloma: Group I—1 eye
**Moustafa K Nassar, Hany A Khairy** **(2018)** [38]	100	LCAG vs. BS + intraoperative 0.02% MMC	12	L: no data M: no data S: no data	Gutt ATB 1 w, gutt AT 1 w, gutt DEX 6 w.	LCAG group—2% MMC group—16%	Graft retraction: LCAG group—2% Fibrosis, symblepharon, conjunctival cyst and dellen were reported only in MMC group—1 eye each Corneal epithelial defect: LCAG group—10 eyes MMC group—12 eyes
**Alvin L Young,** **Ka Wai Kam** **(2019)** [39]	40	LCAG vs. BS + intraoperative 0.02% MMC vs. LCAG + intraoperative 0.02% MMC	180	L: no data M: no data S: no data	Ung ATB tid 4 w, gutt PRED qid 4 w.	LCAG—5.9% BS + MMC—0% LCAG + MMC—0%	Avascular bed: MMC group—3 eyes
**Jing Yu,** **Jun Feng** **(2021)** [40]	85	CAU vs. AMT + IFN alfa-2b vs. mCAU + AMT + IFN alfa-2b	12	L: no data M: CAU:20%/53.3%/26.7% AMT + IFN:16%/64%/20% mCAU + AMT + IFN: 16.7%/56.7%26.7% S: CAU:3.0 ± 1.0 AMT + IFN: 3.0 ± 1.0 mCAU + AMT + IFN: 3.0 ± 1.0	Gutt ATB qid 2 w; gutt AT qid 2 w, gutt FML 3 m (qid→od) AMT + IFN, mCAU + AMT + IFN: gutt IFN alfa-2b: tid 3 m. Sutures removal: YES (14 d)	Conjunctival recurrence rate: CAU: 6.7% AMT + IFN: 12.0% mCAU + AMT + IFN: 6.7% Corneal recurrence rate: 0% in all groups	No complications
**Waleed Alsarhani, Saeed Alshahrani** **(2021)** [41]	94	CAU vs. CAU + intraoperative 0.02% MMC vs. AMT	14	L: 92.6% N/7.4% T M: no data S: no data	Ung ATB + DEX bid 2 w, gutt DEX 4 w.	CAU—15.6% CAU + MMC—15.8% AMT—27%	No data

^1^ Localization (L), nasal (N), temporal (T), double (D), morphology according to Tan scale: T1 (atrophic)/T2 (intermediate)/T3 (fleshy), size (S)—horizontal diameter in mm, mean ± SD. ^2^ Recurrence rate at the end of follow-up period. ATB—antibiotic, BEV—bevacizumab, bid—twice a day, BS—bare sclera technique, BTM—betamethasone, CYP—cyclosporine, d—days, DEX—dexamethasone, FML—fluorometholone, GCs—glucocorticosteroids, Gutt—eye drops, mCAU—modified CAU, IFN—interferon, MMC—mitomycin C, od—once a day, POV—povidone artificial tears, PRED—prednisolone, PTK—phototherapeutic keratectomy, qid—4 times a day, subconj.—subconjunctival, tid—3 times a day, TMC—triamcinolone, ung—ointment, w—weeks.

## Data Availability

All materials and information will be available upon a request sent to the corresponding author by e-mail.

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
