# Peer review of "Evaluating the Efficacy and Safety of Different Pterygium Surgeries: A Review of the Literature"

_ijerph, 2022, doi:10.3390/ijerph191811357_

Round 1

Reviewer 1 Report

I appreciate the authors' effort in analyzing the most important studies on the effectiveness of surgical treatment in pterygium. In the management of pterygium, we can consider three defining aspects;

1. The initial condition of the ocular surface and the appearance of the pterygium

2. The modality of surgical treatment adapted to condition 1.

3. Improving the condition of the ocular surface after surgical treatment and monitoring its evolution.

The article presented moment 2 but its results are closely related to moments 1 and 3.

Each of the surgical modalities presented could be the best choice, provided we identify the situation in which a particular surgical approach

is a minimal and effective therapy.

Even the most complex methods of treatment can fail without improving the local conditions of the ocular surface postoperatively.

A  slightly more nuanced analysis of the results from the presented studies, respectively of their corroboration with the preoperative conditions and the postoperative management of the ocular surface would allow the individualization of the therapeutic modality in relation to the degree of damage of the local tissues and avoid the use of more complex surgeries than would be necessary and implicitly higher risks (e.g. Mitomycin ,5FU) Local preoperative conditions and postoperative management of ocular surface disorders can decisively influence the evolution of the pterygium surgery results and can sometimes overshadow the actual results of the surgical technique used. Pterygium surgery restores the anatomy of a modified and macroscopically visible area, but the functionality of the ocular surface remains altered. Monitoring the ocular surface from tear film to inflammation while avoiding further UV exposure may improve the prognosis of pterygium surgery. An analysis of the presented studies taking into account the preoperative conditions and the postoperative management of the entire ocular surface would clarify the indications of certain surgical and post-surgical approaches.

Author Response

Reviewer 1

We are grateful to the reviewers for their insightful comments on our
paper. We have been able to incorporate changes to reflect most of the suggestions provided by the reviewers.

An analysis of the presented studies taking into account the preoperative conditions and the postoperative management of the entire ocular surface would clarify the indications of certain surgical and post-surgical approaches.

So as not to make the text confusing we put the data about appearance of the pterygium and post-surgical managment in the table.

Reviewer 2 Report

In this manuscript, the authors present important methodological and presentational deficiencies that make it difficult to adequately assess the manuscript.

method: Poorly defined

-inclusion criteria. For example, because no control cases and cohorts have been included in the study, there is a specific reason that should be justified.   What do the authors mean by follow-up in a cross-sectional study?  What do they mean by partial results?

-Exclusion criteria : It is not known for which of the exclusion reasons 824 papers have been rejected.

-Quality assessment method (What method did GRADE use?

The table 1  presentation isis very difficult to follow. In the headings the conclusions are results The authors should present individual tables for each group of surgery and/or treatment and then be able to compare.

They should make tables that specifically contemplate the efficacy and safety objectives. 

Author Response

Reviewer 2

Comment 1-

Inclusion criteria. For example, because no control cases and cohorts have been included in the study, there is a specific reason that should be justified.  What do the authors mean by follow-up in a cross-sectional study?  What do they mean by partial results?

In our paper we focused on clinical trial and randomized controlled trial because they deliver the highest level of evidence, due to their potential to limit all sorts of bias.

Including cross sectional study in our inclusion criteria is a copy paste mistake

We did not include papers with partial results (not finished studies)

Comment 2 –

Exclusion criteria: It is not known for which of the exclusion reasons 824 papers have been rejected.

For our work, we selected publications published from 2008 to 2021 with at least a 6-month follow-up period. The other publications did not comply the above criteria. We focused on the most frequently performed procedures.

Comment 3 –

Quality assessment method (What method did GRADE use?

Authors independently assessed the included studies for potential sources of bias according to Cochrane guidelines. We extracted data from the included studies regarding trial characteristics, methods, interventions, and outcomes.

Comment 4

The table 1 presentation is very difficult to follow. In the headings the conclusions are results. The authors should present individual tables for each group of surgery and/or treatment and then be able to compare.

They should make tables that specifically contemplate the efficacy and safety objectives.

We reorganized the table by adding a column that assessed the effectiveness and safety in the form of postoperative side effects. We also added the column with follow-up period in selected studies.

Reviewer 3 Report

General comments:

It is a very interesting review project that will inspire the reader to rethink their treatment method. The paper also has certain weak points, but these can be easily remedied. Overall, I congratulate the authors.

Specific comments

Line 37: It is true that the main risk factor for the development of pterygium is UV light. Here, much more precise and detailed explanations would have to follow. It is mainly the blue component of sunlight around 300 nm that is discussed as the cause in terms of a dysfunction of the limbal stem cells.  Another important point is that the pterygium occurs predominantly nasally. This should also be mentioned with reference to the literature. This leads to the further question of why it is predominantly nasal? This is largely due to the Coroneo effect and the deflection of the blue component of the sunlight at the cornea in the sense of a prism effect. These important review aspects can easily be found in the literature and incorporated into the introduction accordingly. As a consequence, sunglasses alone are not prophylaxis but sunglasses that are closed temporal and absorb the blue part of the sunlight.

Line 118: MMC is injected subconjunctivally 24 h before surgery. What concentration was used? Applying MMC without rinsing it out can penetrate deep into the tissue and cause serious side effects. What complications were reported?

Line 123: With increasing application of MMC drops lead to more frequent complications is written here. Can this be quantified here? How long were the drops applied and in what concentration? What complications have been described? Is there a dependence between the concentration of MMC and the severity of the complications?

Line 126: What was the complication rate with 5-FU? Where were the granulomas? Subconjunctival? Or were they in the area of the surgical field? Were Vicryl sutures used in the paper cited?

Line 145-148: Beta radiation is a very effective adjuvant treatment method. However, the higher the Sr-90/Ytt-90 single dose and the less fractionated it is, the higher the recurrence and complication rate. A single dose of 2500 cGy is very high. There are many examples in the literature where a lower single dose and higher fractionation are described as having a lower recurrence rate and low complication rate. I recommend finding the papers and citing them accordingly and replacing the one listed.

Line 170: It has been described in a regrowth of the pterygium that it grows less than 2 mm. Is there an interpretation in the paper as to why this is the case and the recurrence is no longer growing? If so, it should be listed here.

Line 193: The most important thing about the surgical result is the lowest possible recurrence and complication rate.

Line 204: How does the cited paper explain the high range of recurrence rates after AMT? This should be addressed.

Line 218: The number of complications seems to be quite high with AMT. Which sutures were used for suturing?

 Line 248: The discussion largely consists of the citations of the literature. More interpretations of the literature should be discussed. In the results, for example, a high complication rate was described for the AMT method. In the discussion, however, a low one was attested. One should merge the discrepancy of the statements.

One very important point was not even mentioned in the discussion. Beta radiation as an efficient adjuvant application should definitely be discussed in detail. This should be done by referring to the literature, where different treatment tactics and the variability of the radiation dose and fractions are published (as already mentioned in the results). This will round off and complete the paper.

Author Response

Reviewer 3

Comment 1

General comments:

It is a very interesting review project that will inspire the reader to rethink their treatment method. The paper also has certain weak points, but these can be easily remedied. Overall, I congratulate the authors.

Response

Thank you very much! We appreciate it.

Comment 2

Specific comments

Line 37: It is true that the main risk factor for the development of pterygium is UV light. Here, much more precise and detailed explanations would have to follow. It is mainly the blue component of sunlight around 300 nm that is discussed as the cause in terms of a dysfunction of the limbal stem cells.  Another important point is that the pterygium occurs predominantly nasally. This should also be mentioned with reference to the literature. This leads to the further question of why it is predominantly nasal? This is largely due to the Coroneo effect and the deflection of the blue component of the sunlight at the cornea in the sense of a prism effect. These important review aspects can easily be found in the literature and incorporated into the introduction accordingly. As a consequence, sunglasses alone are not prophylaxis but sunglasses that are closed temporal and absorb the blue part of the sunlight.

Response

Thank you for pointing this issue. We have added as foloows: “It is mainly the blue component of sunlight around 300 nm that is discussed as the cause in terms of a dysfunction of the limbal stem cells.  Another important point is that the pterygium occurs predominantly nasally. This is largely due to the Coroneo effect and the deflection of the blue component of the sunlight at the cornea in the sense of a prism effect. As a consequence, sunglasses alone are not prophylaxis but sunglasses that are closed temporal and absorb the blue part of the sunlight.XX “(lines- 39-43) and we have added the new reference to the references section:

Murube J. Pterygium: evolution of medical and surgical treatments. Ocul Surf. 2008 Oct;6(4):155-61. doi: 10.1016/s1542-0124(12)70176-3. PMID: 18827948._

Comment

Line 118: MMC is injected subconjunctivally 24 h before surgery. What concentration was used? Applying MMC without rinsing it out can penetrate deep into the tissue and cause serious side effects. What complications were reported?

Response

Thank you for this remark. We have added the following clarification

“In study of Zaky et all74 0.1 ml of 0.15 mg/ml mitomycin C has been injected subconjunctivally into the head of the pterygium one day before surgical excision using the bare sclera technique. Intraoperative eye irrigation with 200 mL of balanced salt solution was done following pterygium excision to wash out residual subconjunctival MMC. As regards postoperative complications, delayed epithelization (more than two weeks) occurred in two eyes (8%) in the injection group and in one eye (4%) in the group with topical application of MMC. Scleral thinning was reported in one eye (4%) in the latter group which occurred at one month and resolved within three weeks under conservative treatment with topical lubricant therapy; no other serious postoperative complications were reported” (lines 124-130)

Comment

Line 123: With increasing application of MMC drops lead to more frequent complications is written here. Can this be quantified here? How long were the drops applied and in what concentration? What complications have been described? Is there a dependence between the concentration of MMC and the severity of the complications?

Response

Thank you for this constructive comment. We have added as follow for clarification sake : “In the study of Hwang et all22 patients were randomly separated into four adjuvant therapy groups, as follows: artificial eye drops four times a day for three months after surgery (control group), 0.02% topical mitomycin C four times a day for five days after surgery, topical 0.05% cyclosporine four times a day for three months after surgery, or topical 2.5% bevacizumab instilled four times a day for three months after surgery. Authors reported that topical MMC and topical cyclosporine treatments had significantly lower recurrence rates compared to the control and bevacizumab groups.” (lines 134-139)

Comment

Line 126: What was the complication rate with 5-FU? Where were the granulomas? Subconjunctival? Or were they in the area of the surgical field? Were Vicryl sutures used in the paper cited?

Response

We have added the following clarification:

“Postoperative complications observed among the studied eyes included granuloma formation 11.4% for 5-FU and 3.0% for autograft and surface infection 5.7% for 5 FU only. The authors have used the cut conjunctiva ends were anchored to sclera with 8–0 silk suture”. (Lines:146-150)

Comment

Line 145-148: Beta radiation is a very effective adjuvant treatment method. However, the higher the Sr-90/Ytt-90 single dose and the less fractionated it is, the higher the recurrence and complication rate. A single dose of 2500 cGy is very high.

Comment

There are many examples in the literature where a lower single dose and higher fractionation are described as having a lower recurrence rate and low complication rate. I recommend finding the papers and citing them accordingly and replacing the one listed.

Response

Thank you very much for this comment. We have added an additional clarification:

Another important issue in the case of success rate is the use of beta radiation (RT) as an efficient adjuvant application. It prevents revascularisation in the mechanism of damaging the endothelial cells lining the lumen of capillaries. Postoperative RT is an efficacious adjuvant therapy used to prevent pterygium recurrence and has been shown to significantly inhibit pterygium regrowth. However, a consensus has not yet been reached regarding the optimal b-radiation dose for pterygium recurrence. Although it has been suggested that 30–50 Gy is the required dose range from a radiation oncology viewpoint (4), a high incidence of scleromalacia has been reported after b-RT. Beta therapy alone produces a significant reduction of the size of the pterygium.25,28,29 However, this is usually insufficient for achieving a complete cure, maybe because the irradiation acts on relatively mature tissues, which are less affected by the irradiation. For better results application of beta therapy after surgery should be applied. The procedure can be performed hours to one or a few days after surgery and it can be repeated within 24 h after surgery, and the at 1 week postoperatively.

Different doses of β-Radiation may cause different success rate. In the study of Yamada et all β-Radiation at 40 Gy was more efficacious than at 20 Gy in preventing pterygium recurrence without scleromalacia development, particularly for large-size pterygia and those in young patients. (lines 327-343)

Comment

Line 170: It has been described in a regrowth of the pterygium that it grows less than 2 mm. Is there an interpretation in the paper as to why this is the case and the recurrence is no longer growing? If so, it should be listed here.

Response

In the paper of Hall et all authors stated :” . The recurrence in the suture group was 1.4 mm onto the cornea in one patient and 2 mm for the other. It both cases the cosmetic appearance and absence of symptoms did not necessitate further surgery”. In our opinion the authors confirmed that there were the cases of regrooving however they did not applied surgical treatment in those cases. We have rephrased these statement accordingly (lines 194-196).

Comment

Line 193: The most important thing about the surgical result is the lowest possible recurrence and complication rate.

Response

We agree with the Reviewer!

Comment

Line 204: How does the cited paper explain the high range of recurrence rates after AMT? This should be addressed.

Response

For the sake of clarification we have added the following answer:

Such difference in recurrence rate depends on various factors, such as differences in the surgical procedures performed (such as the use of fibrin glue and sutureless procedures, the characteristics of the study population (such as sex distribution and mean age of participants) moreover due to the differences in the definition of recurrence as different applications in each surgical method have been suggested. (lines 229-234)

Comment Comment

Line 218: The number of complications seems to be quite high with AMT. Which sutures were used for suturing?

Response

In the paper of Akarbi et al the amniotic graft was sutured by Nylon 10.0 to episcleral.

 Comment

Line 248: The discussion largely consists of the citations of the literature. More interpretations of the literature should be discussed. In the results, for example, a high complication rate was described for the AMT method. In the discussion, however, a low one was attested. One should merge the discrepancy of the statements.

Response

We are very sorry for the lack of clarity. For the clarification sake we have added as follows:

Complications of AMT appear very rare, eventually if they occur, they include: surgery include double vision, granuloma, ocular motility disturbances, and symblepharon. (lines 248-249)

Comment

One very important point was not even mentioned in the discussion. Beta radiation as an efficient adjuvant application should definitely be discussed in detail. This should be done by referring to the literature, where different treatment tactics and the variability of the radiation dose and fractions are published (as already mentioned in the results). This will round off and complete the paper

Response

We are very grateful the Reviewer for this remark. We have added following statements as well we added the appropriate reference too the reference section.

Another important issue in the case of success rate is the use of beta radiation (RT) as an efficient adjuvant application. It prevents revascularisation in the mechanism of damaging the endothelial cells lining the lumen of capillaries. Postoperative RT is an efficacious adjuvant therapy used to prevent pterygium recurrence and has been shown to significantly inhibit pterygium regrowth. However, a consensus has not yet been reached regarding the optimal b-radiation dose for pterygium recurrence. Although it has been suggested that 30–50 Gy is the required dose range from a radiation oncology viewpoint (4), a high incidence of scleromalacia has been reported after b-RT. Beta therapy alone produces a significant reduction of the size of the pterygium.25,28,29 However, this is usually insufficient for achieving a complete cure, maybe because the irradiation acts on relatively mature tissues, which are less affected by the irradiation. For better results application of beta therapy after surgery should be applied. The procedure can be performed hours to one or a few days after surgery and it can be repeated within 24 h after surgery, and the at 1 week postoperatively.

Different doses of β-Radiation may cause different success rate. In the study of Yamada et all β-Radiation at 40 Gy was more efficacious than at 20 Gy in preventing pterygium recurrence without scleromalacia development, particularly for large-size pterygia and those in young patients.

(lines 326-343)

Walter WL. Another look at pterygium surgery with postoperative beta radiation. Ophthalmic Plast Reconstr Surg. 1994 Dec;10(4):247-52. doi: 10.1097/00002341-199412000-00004. PMID: 7865444.

Yamada T, Mochizuki H, Ue T, Kiuchi Y, Takahashi Y, Oinaka M. Comparative study of different β-radiation doses for preventing pterygium recurrence. Int J Radiat Oncol Biol Phys. 2011 Dec 1;81(5):1394-8. doi: 10.1016/j.ijrobp.2010.07.1983. Epub 2010 Oct 1. PMID: 20889266.

Murube J. Pterygium: its treatment with beta therapy. Ocul Surf. 2009 Jan;7(1):3-9. doi: 10.1016/s1542-0124(12)70287-2. PMID: 19214348.